# Adaptive Reservation of Network Resources According to Video Classification Scenes

**DOI:** 10.3390/s21061949

**Published:** 2021-03-10

**Authors:** Lukas Sevcik, Miroslav Voznak

**Affiliations:** 1Department of Telecommunications, VSB-Technical University of Ostrava, 708 00 Ostrava, Czech Republic; lukas.sevcik@uniza.sk; 2Department of Multimedia and Information-Communication Technology, University of Zilina, Univerzitna 1, 010 26 Zilina, Slovakia

**Keywords:** bitrate, subjective video quality assessment, objective video quality assessment, spatial information, temporal information, H.264/AVC, H.265/HEVC, neural network

## Abstract

Video quality evaluation needs a combined approach that includes subjective and objective metrics, testing, and monitoring of the network. This paper deals with the novel approach of mapping quality of service (QoS) to quality of experience (QoE) using QoE metrics to determine user satisfaction limits, and applying QoS tools to provide the minimum QoE expected by users. Our aim was to connect objective estimations of video quality with the subjective estimations. A comprehensive tool for the estimation of the subjective evaluation is proposed. This new idea is based on the evaluation and marking of video sequences using the sentinel flag derived from spatial information (SI) and temporal information (TI) in individual video frames. The authors of this paper created a video database for quality evaluation, and derived SI and TI from each video sequence for classifying the scenes. Video scenes from the database were evaluated by objective and subjective assessment. Based on the results, a new model for prediction of subjective quality is defined and presented in this paper. This quality is predicted using an artificial neural network based on the objective evaluation and the type of video sequences defined by qualitative parameters such as resolution, compression standard, and bitstream. Furthermore, the authors created an optimum mapping function to define the threshold for the variable bitrate setting based on the flag in the video, determining the type of scene in the proposed model. This function allows one to allocate a bitrate dynamically for a particular segment of the scene and maintains the desired quality. Our proposed model can help video service providers with the increasing the comfort of the end users. The variable bitstream ensures consistent video quality and customer satisfaction, while network resources are used effectively. The proposed model can also predict the appropriate bitrate based on the required quality of video sequences, defined using either objective or subjective assessment.

## 1. Introduction

The growing interest in real-time services, especially video transmission over Internet Protocol (IP) packet networks, prompts analyses of these services and their behavior. The interest in these networks is intensifying. Video is a major component of all data traffic via IP networks, and its quality has a key role in multimedia technologies [1]. That was the reason and motivation for selecting this topic for this paper. Maximum perceived quality with effective use of a network’s resources is a challenge to achieve by setting the compression parameters optimally. The assignment of bitrate boundaries for different types of video sequences in terms of dynamics and lighting is described. Video surveillance and data collection from IP cameras for home and retail security solutions have grown over the last few years. Intelligent transport systems, body-worn cameras, and many others contribute to the amount of data. They need to be processed as efficiently as possible in terms of video compression by maintaining the highest possible quality with minimal network resource use. The reputation of artificial intelligence for video monitoring, analyzing video content, and detecting incidents is good. We have found many applications that levy extreme demands and require network resources to be booked on the transfer path for mobile and dynamic wireless networks, such as [2], an example being remote medical procedures. As an artistic endeavor, such a network could be thought of as a concert performed by orchestra members who are in different locations. The IP network architecture was not designed for real-time services or the transmission of data that are sensitive to delay. Many factors can affect the quality of service, especially packet loss, delay, jitter, and bandwidth [3]. According to [4], the proportion of UHD flat-panel TVs will be around 66% in 2023, up from 33% in 2018. The paper aims to define the threshold between the quality of the video and the satisfaction of the end users. A service provider wants to deliver services with the minimum investment possible; implementing quality of service (QoS) tools or quality of experience (QoE) monitoring is a source of costs. However, the customers are not satisfied with the delivery of content of a quality that does not allow them to enjoy the service unobstructed. Users expect guaranteed quality from subscription services regardless of the type of transfer and technology used, and regardless of the network architecture (e.g., centralized or decentralized). Increased user demands and high-quality service expectations have encouraged operators to improve their systems. This need is also stimulated by ultra-high video resolution (4K/8K). In such a flexible and agile background, it is essential to deal with a new model or approaches (e.g., some metrics evaluating user satisfaction, or setting a suitable bitrate to match with compression standard and quality). Our approach uses QoE metrics to determine user satisfaction limits and QoS tools that guarantee the minimum QoE expected by users. It is necessary to evaluate the performance of systems that send information from one source to another (data link), whether that information transfer is efficient and effective. We rated the user perception of transmission quality against the quality of its content for end-to-end delivery systems (IPTV). Generally, it is assumed that high performance and transmission quality leads to excellent user satisfaction with services. The proposed model brings the advantage of a variable bitrate that changes by type of the scene and simulations of the subjective quality of the scene using a classifier based on the artificial intelligence network that was published in [5]. It keeps a constant quality for the end users and also safe bandwidth for the providers. In the research community, some surveys about adaptive streaming [6,7] and prediction of the quality of the video (especially QoE) [8,9,10] exist. The term QoE originated as a counterbalance to QoS and included something that addresses human perception and experience, as these were considered more appropriate for designing highly acceptable systems and services. QoE reflects the fact that perceived quality is a crucial factor in evaluating structures, services, and applications. To this end, research is continuing into finding networks and various support technologies that can evaluate QoE. It is necessary to explore possibilities that will increase network quality, user satisfaction, and save network resources.

A lot of papers deal with the topic of video quality. We describe models, procedures, and analyses of the related research in Section 3. Quality evaluation is described in detail, but we could not find a model defining the relationships between perceived quality and mathematically calculated subjective metrics, or some tool for predicting subjective quality based on objective assessment. We implemented a tool for retaining perceived quality using a variable bitrate for different types of video sequences. Our model offers a prediction of subjective quality, and a mapping function that compares subjective and objective assessments.

The evaluation results led us to set the appropriate bitstream based on the compression standards, resolution, and scene type used. Our model gives recommendations for selecting a suitable bitrate for individual video scenes to achieve subjective quality.

## 2. Scope and Contributions

In summary, this paper has the following novel contributions:Creating a database of video sequences (Section 5.1).Video sequence classification by spatial information (SI) and temporal information (TI) (Section 5.2).Evaluation of the video scenes by objective and subjective metrics (Section 5.4).The proposal of a new classifier based on artificial intelligence (Section 6).Determining the correlations of the results for the objective and subjective evaluations of the video (Section 7), followed by a prediction of subjective quality.Creating an optimal mapping function for predicting objective and subjective evaluations of the quality (Section 8).−Prediction of bitrate based on requested quality estimated using the SSIM index.−Prediction of bitrate based on requested quality using MOS.

The rest of the paper is organized as follows, Section 3 reviews the state-of-the-art in QoE and adaptive streaming. The aims of the paper are presented in Section 4. Section 5 provides insights into the data preprocessing, whilst Section 6 explains proposed classifier and neural network. Selection of a suitable topology is explained in Section 7, the development of the prediction model is described in Section 8, and the results of validation experiments are presented in Section 9. Finally, Section 10 concludes the paper.

## 3. Related Work

The papers that deal with exploring video quality are described in this part. The perceptions of the human brain cannot be replaced by computing through algorithms. A more reliable method is assessment by end users. Quality can be defined as the degree to which something meets a set of requirements. For video, it is determined as a result of perception by the human visual system.

The quality of video sequences can be adapted based on the customers’ requirements and the results of their perceptions of quality. Thus, providers can change settings regarding quality, such as the bitrate of a scene. From this point of view, both QoS and QoE are essential to video content providers’ considerations. They are essential to the satisfaction of the end users, as they lead to an increase in the enjoyment of these services. QoE is detailed in [11].

The authors of [7] performed a survey for QoE, which was structured according to the influence of video adaptation on QoE. This research presented an overview of the actual state of affairs and recent developments. The authors also described methods of HTTP video streaming and the point of view of the end users evaluating the video. VQAMap was introduced as a new tool for the mapping of objective evaluation to the subjective MOS values in the paper [12]. This mechanism uses public databases of the VQA evaluation. They also evaluated the accuracy of quality estimation and compared the power deviation of the objective metrics. The quality of the approximation was assessed using a correlation, and the accuracy of the quality estimate was around 95 percent. Data from three public databases were selected for the model. Subjective data were converted to the same input scale (according to the selected MOS scale, for example, 1–5). The model uses nonlinear regression and inverse interpolation. The model evaluates lower-resolution videos using the H.264 compression standard.

The authors of [13] created a new no-reference metric for video quality assessment based on the bitrate. Firstly, it checks the influence of the bitstream, and the model was improved through features for visual perception such as TI, image contrast, and texture. The final model was established using the appropriate settings of the weights for the parameters. Ten types of reference scenes and a database of 150 test samples were used for testing of the metric. The Spearman and Pearson correlation coefficients verified the effectiveness of the model.

The framework for streaming video over DASH was developed in [14]. Optimal bitrate switching was proposed for the satisfaction of the end customers. The aim of the model was also to maximize the overall performance in the created framework.

The paper [15] analyzed the impacts of the compression and network distortion on streaming video. The authors proposed a reduced-reference metric for perceptual video quality evaluation using optical flow functions. The deviation of the damaged frames is used in this proposal. The streaming artifact and distortion sources are taken into account. The distorted video sequences from three databases for the video quality assessment were used for the validation of the proposed metric. With this approach it is possible to predict the perception of quality. Results expressed by the Pearson and Spearman correlation coefficients were evaluated for the artifacts of compression, channel-induced distortion, and other distortion.

The authors of [16] dealt with HTTP adaptive streaming. They focused on longer sequences with durations of approximately two minutes for the evaluation of the video quality. Six video sequences in FHD resolution with a framerate of 24 fps encoded by H.264, characterized by the SI and TI, were used. The paper examined the influence of adaptive bitstream modifications during streaming.

Adaptive HTTP streaming using driven rate adaptation is introduced in [17], which is appropriate for providers and for customers. The authors computed the initial delay—interruptions in the transmission—using the proposed algorithm. Based on it, they adapted the bitstream of the streaming video dynamically for maximum customer satisfaction. The main idea of the paper was to satisfy QoE for the streaming videos. The authors evaluated the subjective and also the objective quality of the test sequences. Spatial and temporal information characterize the video sequences. The paper [18] dealt with 5G network architecture with satellite communication for the achievement of QoE in HTTP live streaming. The authors proposed a network with a mobile and satellite network operator and a video provider.

A model for the extraction of the multifunction elements in the spatial and frequency fields was deployed in the paper [19]. It uses the proposed filter log-Gabor for visual information. They used spatial frequency for the variations of the sensitivity to contrast and texture. A new model for QoE evaluation of the new and advanced multicast services for the 5G networks was deployed [20]. It uses the measure of deformity in the depth map. The relation between the depth map and the quality of the picture is described.

Other authors developed a model that connects different information from videos together in [21]. They presented a stacked auto-encoder, sparse representation, and saliency. On that basis, the authors constructed a model for the combination and analysis of various features.

The paper [22] examined the perceptibility of packet loss deficiency in the scenes’ temporal and spatial sections. End users tapped the touchscreen while the video was streamed, in the locations of artifacts. The authors analyzed objective attributes from these artifacts to create the various models for combining characteristics into a new metric based on the objective evaluation of the artifacts’ noticeability.

A new metric to evaluate the stalling of streaming video without access to a buffer was designed in the paper [23]. The algorithm can automatically detect the quality of freeze frames from decoded video based on their global intensity and the characteristics of the local texture. A linear model is used for quality evaluation. The advantages of the proposed metric are the independence of the data and platform adaptability. The paper [24] describes the advantages of machine learning for the quality prediction of the videos and describes the limitations of the current state. The subjective database for the panoramic scenes was created in the paper [25]. Change of video sampling was suggested before the coding during the subjective assessment. Sixty panoramic videos with coding degradations were used for the subjective evaluation of the quality using the ACR-HR method by thirty evaluators. The well-known support vector machine and Gaussian mixture model and the new convolutional neural network (CNN)-based automatic speaker verification were applied and subjected to synthesized speech in the paper [26]. This work analyzed and evaluated the ability of commonly used ASVs to recognize the user-targeted speech from the synthesized speech.

Other authors published several articles on the issue under consideration. The papers [27,28] describe the simulation of a network traffic and prediction model for the subjective assessment of the video sequences damaged by packet loss. Based on the proposed analytical model, we developed an application to predict QoS qualitative parameters in IP networks concerning the strategies used for packet processing in routers and the capacity of the network. The model can predict the quality of triple-play services according to the transmission speed of the router interface in packet networks.

Paper [29] presented a method of estimating users’ perceived quality of UHD video experience flows in the emerging 5G networks as part of a comprehensive 5G-QoE framework. The authors used the four video sequences from Ultra Video Group and the five from Mitch Martinez. They used SHM 6.1 to encode the video scenes to H.265 compression standards and two resolution types (Full HD and Ultra HD). The results of subjective testing achieved accuracy of up to 94%. Paper [30] presented a simulation of typical and adaptive streaming scenarios for mobile devices. For that reason, Quad HD resolution was used. The author described the perceived video quality for both cases using three types of scenarios with different bandwidths. Forty humans evaluated five different video scenes with the subjective ACR method. Subsequently, the authors compared adaptive and standard streaming. They could not find the complex solution for deploying a variable bitstream based on the scene type.

Our paper presents an optimal mapping function for predicting objective and subjective evaluations of the quality. Our paper’s main benefit is the possibility of setting an appropriate bitrate for the individual video sequences based on a new classifier for predicting the boundaries SI and TI following the quality requirements. Differences between the last two papers and our approach are shown in Table 1.

## 4. The Aim of the Paper

The current section highlights the aims of this paper and further steps to be made in the field of video quality evaluation, namely, research into the practical use of bitstream. The buffer is an essential part of the encoder and decoder. It balances the variable bitrate depending on the content of the current scene and compression applied. A static view of television news or political discussion needs a lower bitrate than a highly-dynamic action scene, an advertising spot, or a dynamic sports match. Sports broadcasting needs to be differentiated into more categories: high motion sports (such as hockey or football), dynamic sports (such as darts), and nearly static sports (such as chess). Therefore, the purpose of this work was to determine the proper bitrate for specific scenes and achieve the highest end user satisfaction regarding quality. One of the aims was to define boundaries for individual sequences based on the spatial and temporal content. A suitable bitstream could be assigned for each resolution and for each boundary. It is possible to save a significant portion of the average bitstream by deploying a variable bitrate (VBR). The average and the maximum bitstream values of VBR are important. One needs to select the appropriate test scenes to determine video quality and encoding.

The variety of dynamics plays a primary role in the selection of video scenes. If the quality evaluation results are to be trusted, a big database of the samples is indispensable. Twelve of them were selected by the content complexity characterized by SI and TI in our experiment. The authors used objective and subjective metrics for the evaluation of the video sequences. The subjective estimation of the video followed ITU recommendations P.910 [31] and ITU-T BT.500-14 [32]. People of both genders of various ages evaluated the video sequences. Both approaches of video evaluation defined the correlation relationship. The prediction of the subjective assessment based on the objective uses this continuity.

The main aim of this paper study to propose a new classifier for predicting the boundaries using SI and TI, while following quality requirements. Based on these constraints, an appropriate bitrate could be set. Content classification analysis is performed by calculating the time and spatial information indices relative to the luminance component of each video. We define the boundaries for time and spatial information using the mapping feature. Thresholding and scalability for bitstreams are used in the model for adaptive conservation of network resources according to established boundaries. We introduce a new classifier that determines the scene type based on TI and SI information. Depending on the scene type, the model determines the appropriate bitrate based on a mapping function. The results from the model were verified by comparing predicted video quality using the proposed classifier with the required bitrate value. It is expected that the numerical expression of quality obtained using the objective method would be better correlated with human subjectivity. Various metrics can express the correlation between subjective and objective evaluation. The two most common statistical parameters related to expressing performance are root mean square error (RMSE) and Pearson’s correlation coefficient.

Our classifier’s advantage is the knowledge that the image quality is insufficient, sufficient, or perfect. If a video with specific parameters achieves the best quality observable by the human eye, it is unnecessary to use better parameters because the eyes do not notice them. Thus, the service providers can know the scenes’ qualitative settings needed to achieve maximum customer satisfaction. It can dynamically change the quality of various video sequences and offer the full degree of satisfaction during the whole broadcast. Examples of usage are online broadcasting and streaming sessions. When a broadcast from a studio is alternated with a dynamic scene, it is possible to change the scenes’ qualitative parameters. The advantage is that the videos will have a different quality based on the content; qualitative parameters for the studio broadcast with slightly dynamic movements can be lower. Another example of usage is virtual reality, where the quality of processing and of the video scenes affects the resulting experience.

## 5. Data Preprocessing

Some partial steps are necessary to do for designing of the classifier based on artificial intelligence. The first is creating a video database distinguished by the SI and TI encoded to require a format of specified resolution, compression standard, and bitstream rate; evaluation of the video quality and classification of the video sequences are described in this section.

### 5.1. Creating the Database of Video Sequences

A testing database of the video scenes was prepared. Researchers from the Media Lab of the Shanghai Jiao Tong University [33] dedicate their work to video signals. The sequences in their laboratory include a range of topics related to the video signal. They are working on video analysis, processing, and compression. A set of fifteen 4k video test sequences for downloading in the uncompressed format—YUV 4:4:4 color sampling, 10 bits per sample with a frame rate 30—are available for download. Our database needs to cover a wide range of video scenes in terms of SI and TI. After calculating these parameters for each sequence from the Media Lab, we chose eight of them with various pieces of SI and TI. These broadly cover all quadrants from TI and SI viewpoints. The Ultra Video Group (UVG) of the Laboratory of Pervasive Computing in the Tampere University of Technology [34] is a leading academic video coding group in Finland with many years of experience in this area. We used four 4K test sequences made available for download in raw format as complementary video scenes to our database. These suitably expanded the SI and TI distributions in the less covered quadrants of the graphs of SI and TI dependence, especially for video scene “ReadySteadyGo”. These four additional video sequences were used to expand the database for objective evaluation and bitstream prediction based on an objective SSIM metric. Hence, twelve source video sequences (SRCs) were used in our paper. SI and TI can be compared in Figures 4–7. From these graphs it can be seen that the sequence selection covers a variety of scenes.

The neural network simulates individual sequences encoded by different values of the bitstream, compression standard, and resolution. SI, TI, and quality parameters characterize the video scenes. Individual video scenes are varied by the speed of the movement, object movement, details of the frames, and light intensity. Scene selection was performed by ITU-T Recommendations P.910 [31] and ITU-T BT.500-14 [32].

The original samples from the database were created as ten second video sequences encoded with 30 frames per second, Ultra HD resolution, 4:4:4 chroma subsampling, and ten bits of the bit depth per channel. Video scenes longer than ten seconds do not influence the quality evaluation. We had to convert all SRCs to YUV 4:4:2 typical chroma subsampled, using FFmpeg [35] as a first step. The encoding was repeated for all SRCs to convert them to Full HD resolution, two types of codecs, and eight types of a bitrate. Encoding parameters are shown in Table 2.

Thus, each SRC for both resolutions was encoded to H.264 (AVC) and H.265 (HEVC) compression standards, and eight different bitstreams (Figure 1).

### 5.2. Video Sequence Classification

Spatial and temporal information were used for identification of the video sequences. These values classify the video content of each scene in our database, and to have a different rate for each video sequence encoded to a different resolution, bitrate, or compression standard. Numerous types of video sequences with different SI and TI were adopted. We complied with the ITU’s recommendations [31,32] for conducting subjective tests. The MOS scale was applied for subjective video evaluation. Generally, spatial and temporal information is used to represent video content. This information is the critical and key variable of a given scene and determines the compression rate of the video. The content classification was analyzed by computing the SI and TI indices for each video sequence.

#### 5.2.1. Spatial Information

Spatial information is based on the Sobel filter (the Sobel operator) that uses the edge of the frames for taking information about frequency which determines SI. A higher value of SI indicates increasing sequence spatiality. First, each video frame is filtered by the Sobel filter at the specified time. The standard deviation ($std_{space}$) of the pixels is then calculated for each filtered frame (see Figure 2).

This operation is repeated for each frame in the video sequence, and the result is the highest value in the time series ($max_{time}$) representing the SI content of the selected scene (Equation (Equation 1)) [16] (parameters are shown in Table 3):(1)$$SI=max_{time}\left[std_{space}\left[Sobel\left(f_{n}\right)\right]\right]$$

#### 5.2.2. Temporal Information

This information indicates changes in the video sequence motion over time. A greater value indicates a higher sequence motion. From a technical point of view, TI is based on the function of movement difference. This feature is a difference between pixel values at the same place in the space in the successive frames of the video (see in Figure 3). This information determines differences in the movement of the frames or consecutive times, which defines the function (n) in Equation (Equation 2):(2)$$M_{n}\left(i,j\right)=F_{n}\left(i,j\right)-F_{\left(n-1\right)}\left(i,j\right)$$
where $F_{n}\left(i,j\right)$ is the pixel of the image F at the $i^{th}$ row, $j^{th}$ column, and $n^{th}$ frame in time. The standard deviation is calculated from the function above, and the maximum value is found by Equation (Equation 3):(3)$$TI=max_{time}\left[std_{space}\left[M_{n}\left(i,j\right)\right]\right]$$

Equation parameters are shown in Table 3.

The goal of the classifier is to determine the boundaries of SI and TI for bitstream switching. No reference Mitsu tool [36] was used to evaluate the level of distortion in the video scenes. We can compare results for H.265 compression standard for both Ultra HD and Full HD resolutions between 3 and 20 Mb/s in Figure 4, Figure 5, Figure 6 and Figure 7. There is information displayed about SI and TI dependence for these videosequences. The paper proves the variability of SI and TI values across the encoded parameters, such as bitrate, resolution, and compression standards. The classification of the video samples can thus be done by SI and TI.

From the graphs, we can see that the created database consists of the various video sequences, thereby covering a wide range of scenes, which makes it possible to adapt the results in practice.

### 5.3. Evaluation Methodology

The data analysis consists of the precise steps: data elimination, accurate statistical assessment, and presentation of the results. Irrelevant data need to be detected, and data out of the confidence interval need to be determined. Those data will not be included in the final presentation. The Pearson correlation coefficient commonly used in linear regression can specify the relevance of data. Spearman’s correlation coefficient can be used as a secondary method to estimate relevance. Thirty out of thirty-five evaluators were included in the evaluation. Ratings from the five respondents were not accepted because their results differed from other values. The average age of participants was thirty-five. Of these, the fifteen men were twenty-one to forty-four, and the fifteen women were nineteen to thirty-four.

### 5.4. Evaluation Dataset

**Subjective assessment:** We followed recommendations in [31,32] that describe conditions for the reference video scenes; instructions on how evaluators can do the evaluation, adjust brightness, adjust distance from the monitor, process, and analyze results; and many other things. The minimum number of evaluators needed to be fifteen. Each of them had the same instructions and conditions for carrying out the evaluation.

The legend for Figure 4, Figure 5, Figure 6 and Figure 7 is displayed in Figure 8.

A video laboratory with a recommended condition was used for the testing. We gave clear instructions for carrying out this work to participants in the evaluation. It included the condition that the video sequences were evaluated in terms of image quality, not scene selection and processing. The demonstration of test sequences was performed. The guidance included the scale of the evaluation, video quality degradation.

We decided on the ACR subjective method, which evaluates the scene’s quality without comparison with the original. It illustrates the conditions in a real deployment, where the end customer has the opportunity to see only the broadcast video. This method is faster when we compare it with DSIS or DSCQS.

**Objective assessment:** In this case, the MSU Video Quality Measurement Tool [37] was selected for evaluation. Reference video scenes were compared with the test samples.

The results of the objective assessment using the SSIM metric for the UVG sequence are shown in Figure 9 for Full HD and in Figure 10 for Ultra HD resolution. The results for the chosen SJTU Media Lab video sequences are in Figures 12–15 in Section 7. The evaluated data are compared with the objective and subjective metric in the selected video.

## 6. The Classifier Based on an Artificial Intelligence

A new classifier for prediction of bitrate and video evaluation is proposed. It knows to predict both subjective and objective approaches using artificial intelligence. MATLAB was used for creating a neural network. The term for this network was derived from the basic functional unit of the human nervous system or nerve cells that are present in the brain and other parts of the human body. The neural network is trained by successively adjusting its weights and biases (these variables are also called free parameters). This procedure is performed iteratively to obtain the desired output. These variables are also called free parameters. Training is needed in the first phase of the neural network (learning). It is implemented using a defined set of rules known as the learning algorithm. This process needs a set of training data, and it is then tested on a set of testing data.

Our model implements a multilayer neural network using back-propagation, which can update weights and reduce an error function to its local minimum by moving weights forward and backward. The model with a feed-forward neural network was implemented in MATLAB R2020b. We used three types of gradient modes (the convergence rate, stability, and complexity of these methods are different):Batch gradient descent (traingd).Batch gradient descent with momentum (traingdm).Gradient descent with variable learning rate (traingdx).

### 6.1. Data Processing

Training of the neural network can be more effective if inputs and outputs are done by scaling in the same interval. We tested scaling at <−1; 1> and <0; 1> intervals (we also tested training without scaling inputs and outputs). Network training was done separately for the H.264 and H.265 codecs and both codecs at once, which made it possible to compare network training for two different input groups.

We trained the neural network ten times for each topology for more accurate results. Weights and biases were always randomly generated. This has limited the impact of randomly generated initialization weights, which has increased the statistical significance of the whole process. A validation set of data was used during training to verify the rightness of the network training to prevent it from being overloaded. Training and testing sets were created from the database. It was a random choice into 75:25 for training:testing. After that, the validation set was generated from the training database at a ratio of 70:30 for the training set. This process was repeated ten times to guarantee statistical significance. Each repetition trained the network with different training and validation sets. The training was performed for multiple topologies that contained two, three, and four layers of concealed neurons, with up to 160 neurons in one layer. The network was overburdened with a higher number of layers; a higher number of neurons in one layer was not necessary, as the network was able to properly train with fewer neurons, which demonstrates the results of the simulation. The ratio of data distribution, number of layers, number of neurons resulting from training, and simulation tests show the results of the final programmed algorithm and the results obtained by looking for the proper functions and selecting the correct data layout. The number of repeats chosen was also based on the results of the authors [38].

We used a validation set to prevent network overreliance. If the error rate of the validation set did not decrease six consecutive times, the training ended. The best results from the training were from the time when the network was best trained (before recording the first occurrence of six error rate of the validation set). As a result, the weights and biases were reached with the minimum error of the validation set. Thus, if the error rate of the training data dropped, but the error rate of the validation set increased or remained at the same level, training stopped. We tested changing the number of incidences of error rate on the validation set, but no better results were achieved. The possibility to modify the error calculation function was also verified. We performed a simulation for both resolutions separately and another simulation together, and for both compression standards. It is necessary to test network training using test data that the neural network (Figure 11) does not yet have available. We calculated the correlation based on the Pearson correlation coefficient after data simulation. If the error rate of the test set is significantly lower than the error rate of the validation set, this would mean a poor partitioning of the video sequence database.

### 6.2. The Classifier

The neuron-based classifier predicts the appropriate bitrate based on the scenes classified by the critical SI and TI and the qualitative parameters of a given video sequence. The model allows the prediction of:Bitrate, which depends on the type of scene and its quality evaluated by an objective SSIM metric.Bitstream based on the kind of sequence and its quality estimated using the MOS scale quality score (value achieved with the subjective ACR metric).

Besides bitstream prediction, the designed model can predict subjective video quality based on the type of the scenes, objective expression by the SSIM metrics and encoded video parameters. In practice, end customer satisfaction is significant, and it is essential to evaluate the perceived quality of the viewed video sequences by users. The objective metric itself does not give us information about the perceived quality of the content being viewed by users.

In addition to bitstream and subjective quality prediction based on the objective evaluation, quality can be predicted only from the video scenes. The model thus can predict the subjective and objective quality of the video sequence identified by the type of it and bitstream.

## 7. Correlation of the Results between the Objective and Subjective Evaluation of the Video

The database with 384 video sequences was created. It includes twelve types of the scenes encoded into two resolutions and compression standards and eight bitstreams. We classified each video from our database using SI and TI information and evaluated quality using objective tests. The subjective metric evaluated SJTU Media Lab Sequences (265 scenes). In this section, the results of the quality of the video sequences from the created database are evaluated. The correlation between subjective and objective quality evaluation and the model of subjective quality prediction based on the objective are also described. The results obtained using the subjective ACR metric as an average MOS for the selected video type can be seen in Figure 12 (Bund Nightscape) and Figure 13 (Wood). These results can be compared with the values obtained by evaluating video quality using the objective SSIM metric in Figure 14 (Bund Nightscape) and Figure 15 (Wood).

From the figures, we can see that subjective metrics are very similar and close to objective metrics. The best score is seen with the H.265 standard in Full HD resolution and the worst score with H.264 in Ultra HD resolution. The best results for Full HD have slow-motion scenes (scenes with lower TI value) such as the “Construction Field” and “Tall Buildings” scenes. The resulting curve of the Wood sequence (sequence with the highest SI and TI values) is located around half of all quality curves. We can say that the human brain does not detect the motion of the camera and the movement of the object on a static background. Minor scoring differences in the MOS scale are at the higher bitrate. Likewise, slow-motion scenes achieved the best results in Ultra HD resolution—in this case, the Construction Field and Bund Nightscape scenes.

The average results show high similarity between the subjective (Figure 16) and objective (Figure 17) tests. The quality of all the analyzed sequences increases logarithmically. Users rated video quality very similarly compared to the SSIM metrics. It confirms the credibility of subjective tests. The human brain does not perceive the change of codec with a high bitstream as intensely as in objective evaluation. In this case, user ratings move roughly to a similar level. The efficiency of coding using H.265 is higher than for H.264; H.265 surpassed H.264 in both Full HD and Ultra HD.

### 7.1. Subjective Quality Prediction

If our model knows quality estimated by objective metrics and the scene’s type, he can predict subjective quality. That is the main topic of this chapter. It focuses on learning topology selection, continuously testing, and simulating topology. Additionally, human video quality comprehension by the objective value with SSIM metric is described.

### 7.2. Selection of a Suitable Topology for the Neural Network

For achieving a trained neural network is necessary to set the right number of hidden layers and neurons in each of them. If the number of neurons is too low, the neural network cannot reach trained condition. Too many of them lead to the overtraining of the network—that means that it is trained only for learned values and cannot to predict a new one. The process of creating the neural network is shown in Figure 18.

Our model works with five inputs entering the system (the type of scene consists of SI and TI, resolution, compression standard, objective evaluation). Based on it, we defined basic topologies for the neural network. The first layer consists of five neurons (five inputs), the last layer has only one neuron (one output), and some hidden layers were also created in primary topologies:5-1;5-1-1;5-3-1;5-3-2-1;5-5-3-1.

For more accurate results, any simulation was repeated ten times. Each time, the simulation starts with randomly generating weights and biases. Simulations were done for zero, one, and two hidden layers. One of the steps in each simulation is to divide a set of data into the training and testing group. After that, allocate validation data from the training collection. It is necessary to note that this process is always randomly generated, and so each simulation brings relevant results. At the moment, our network is trained, the next ten rounds of the simulations with randomly generated training and validation sets are performed. So that for each of the ten network modeling repetitions, a training and validation set were randomly generated ten times (described in Section 6).

Pearson’s correlation coefficient, number of iterations, and time needed to reach them (mean values) for the Ultra HD resolutions are shown in Table 4. When we compared the best results for primarily defined layers, two layers model (without the hidden layer) reached better results. The criteria for selecting the best topologies were based on correlation and the associated error rate followed by the number of neurons and the associated time to get optimal topology.

Subsequently, we selected the two-layers model for the next simulation and looking for even better results. We started with a small number of neurons, which we gradually increased. From the results, we again selected the best simulations. Table 4 shows the results (simulations were also done separately for compression standard H.264 and H.265 for Ultra HD resolution). Mean squared error (MSE) and the Pearson correlation coefficient for each data set of the simulations are displayed in Table 5 (simulations were also done separately for compression standard H.264 and H.265 for Ultra HD resolution).

We compared more algorithms and settings of the simulation. The gradient descent with variable learning rate algorithm (“traingdx”) achieved the best results with the “tansig” activation function, and when data were scaled in the interval <−1; 1>. The algorithm “traingdx” needs fewer iterations for training.

## 8. Creating an Optimal Mapping Function

The mapping function for predicting objective and subjective evaluation of the quality to determine the boundaries of TI and SI was developed. For proper usage of variable bitrate, it is crucial to set it up correctly to achieve optimal quality for each video. It was necessary to select a wide range of video sequences to cover the various scenes for this purpose and to make the model usable. We chose these videos based on the critical parameters which specify their type. The mapping function can evaluate the optimal bitrate to fulfill the specified quality in the form of MOS or SSIM index. The suggested model then estimates how a bitstream meets established quality requirements for the type of video sequence. It also allows the opposite kind of prediction—to evaluate quality based on the type of video sequence and bitstream. It can predict quality in the form of MOS or SSIM value.

### 8.1. Prediction of Bitrate Based on Requested Quality Estimated Using the SSIM Index

The model’s purpose is to define the bitrate for each type of video sequence. In this case, we describe the model for prediction bitstream using the estimated quality represented by the SSIM value. Input variables of the neural network are information about the type of scene identified by SI and TI, qualitative parameters and required quality expressed by SSIM. We compare each resolution separately in one simulation because we would like to know the differences in the model’s ability to predict bitrate in different environments. As technology is continuously evolving, it is important to focus on the assumed use of higher resolution. It is another reason for a separate simulation of compression standards in Ultra HD resolution. It allows us to look at determining the appropriate bitrate of this resolution, whether VBR is deployed, or that the prediction of this model could be used for the minimum bitrate using Constant bitrate (CBR).

Table 6 shows the best simulation topologies from Table 7 for resolutions separately and together. The statistical evaluation of predictions by the model for selected topologies is given in Section 9.2.

### 8.2. Prediction of Bitrate Based on Requested Quality Using MOS

The satisfaction of end customers is the most important factor in evaluating quality. The opinion of the user and his perception of quality is a benefit to the provider of triple-play services. If the provider knows how the customer perceives quality, it can adjust the quality of the offered content and set up the appropriate quality parameters as bitstream. We used different types of video sequences evaluated by users. These scenes were encoded using a variety of bitrates. We have created a comprehensive database of rated videos based on different bitstream changes. It is possible to determine the appropriate bitrate based on the desired quality expressed by the MOS scale and predict it with the proposed model using the database.

In Table 8, we can see the best topologies of bitrate prediction based on information about the scene and evaluated the quality of the video sequences using MOS. The top topologies are selected based on the average Pearson correlation coefficient of the test set, the number of iterations and the associated time to get an accurate topology.

Table 9 contains the results obtained by simulation at selected topologies from Table 8. The smallest achieved error is calculated during training using the MSE function. Correlations are evaluated using the Pearson correlation coefficient for all of the data sets (training, validation, and test) in the simulation. Statistical evaluation of model predictions for selected topologies is given in Section 9.3.

## 9. Model Verification

This section deals with verifying individual predictions and describes how a designated classifier, using artificial intelligence, can predict the subjective video quality evaluation based on objective assessment, the appropriate bitstream based on the required video quality and the qualitative parameters of the scene.

### 9.1. The Prediction of the Subjective Value

This section describes verification and validation of subjective quality prediction based on the scene, bitstream and objective result by SSIM. Topology 51-25 was selected as the best in the model to simulate both compression standards simultaneously (described in Section 7.1. This topology achieved the best correlation and predicted data. In Table 4 we can also see the results of the model prediction for both compression standards used separately and compare them to each other.

The statistical evaluation needs to confirm the simulated results. Several methods for the verification were chosen. A 95% confidence interval was used.

#### 9.1.1. The Descriptive Statistics

It was made between reference and test set for the analysing of data dependability. The minimum and maximum values, average values, medians and confidence intervals are quite similar (it can be seen in Table 10). We can assume that the simulation and developed model are credible from this statistical evaluation because of the data statistics similarity.

#### 9.1.2. The Correlation Diagram and Root Mean Square Error (RMSE)

From the Figure 19 is possible to see the similarity between statistics. It is confirmed by the RMSE, a measure of the deviation between reference and streamed data. It measures the average magnitude of the error. The value of RMSE = 0.066 represents the error rate of 1.32%.

#### 9.1.3. Probability Density Function

The probability density is the relationship between observations and their probability. A probability distribution can be understood as a representation that assigns each elementary phenomenon a real number that characterises the likelihood of this event. The probability density is a function whose value for any selected element from possible patterns (random variable values) can be interpreted as the relative multiplicity of the element value within the entire collection of possible patterns. The Figure 20 show that the simulated set is close to the reference, which again confirms the credibility of the model.

### 9.2. Prediction of Bitrate Based on SI, TI, and Video Quality Characterized by Objective SSIM

This section includes verifying bitstream prediction for the required video quality evaluation using the SSIM metric. An evaluation of both resolutions and compression standards in one run is given.

#### 9.2.1. The Descriptive Statistics

The Table 11 shows the proximity of mean values, medians, and confidence intervals between original and prediction values. A comprehensive video quality assessment tool was developed for different collections of scenes that are coded for different compression standards and resolutions. The bitstream prediction results have a high prediction accuracy; therefore, simulations can combine two resolutions. The model adapts to given requirements and recognises the difference in quality for individual and fundamentally different groups of videos. The first data column in the Table shows results for Ultra HD separately, second data column for Ultra HD and Full HD resolutions.

#### 9.2.2. The Correlation Diagram and RMSE

This simulation has a slightly higher error than the BR prediction based on MOS. RMSE function represents 8.015% of the prediction error for UHD (Figure 21) The correlation diagram and RMSE function for Full HD and Ultra HD in one run is shown in Figure 22. Using RMSE calculations, the estimated error rate is 5.55%.

#### 9.2.3. Probability Density Function

Distributive density trace for reference and simulated sets in BR prediction based on required SSIM quality for Ultra HD resolution is shown in Figure 23. We can compare it with the results for Full HD and Ultra HD resolution in one run in Figure 24. The density trace of these simulations visually confirms the correlation of the test and reference sets.

### 9.3. Prediction of Bitrate Based on SI, TI and Video Quality Characterized by MOS

This section verifies the bitstream prediction results for the input information characterizing the scene and expresses the user’s quality rating of the given video, represented in the mean value of the MOS.

#### 9.3.1. The Descriptive Statistics

The Table 12 shows the similarity of descriptive statistics for bitrate prediction based on MOS for Ultra HD resolution and also for Full HD and Ultra HD resolution in one simulation.

#### 9.3.2. The Correlation Diagram and RMSE

When RMSE is calculated for bitstream prediction based on SI, TI, and MOS, it corresponds to a 3.41% error rate for Ultra HD resolution (see Figure 25) and 3.25% predictive error for both resolutions (Figure 26).

#### 9.3.3. Probability Density Function

Distribution density trace also shows the similarity of reference and test sets for bitrate prediction based on MOS for Ultra HD resolution in Figure 27 and also for Full HD and Ultra HD resolution Figure 28.

## 10. Conclusions

The paper dealt with a variable bitrate for the streaming of video in triple-play services in compliance with QoE. For scene description we applied spatial and temporal information. The bitstream classifier was developed using artificial intelligence. The mapping function that can find the optimal bitrate to fulfill the quality requirements for a variety of video scenes was described. It would allow the essential quality of the video to be transmitted to end users. The advantage of VBR is the efficient use of network resources for consistent video quality.

It is necessary to provide the highest possible quality for broadcast content, as customer satisfaction is the primary goal when providing triple-play services. Our solution offers a model which can be used to determine the qualitative settings to achieve this goal.

The quality was evaluated by both types of metrics (objective by SSIM and subjective by ACR). Based on the described correlation, the MOS prediction of subjective quality can be made.

The generated model evaluates the appropriate bitrate for the desired quality of a given video, defined by characteristic and qualitative parameters. The scene can carry a dynamic flag, or the video sequence can be converted in real-time, which would be time-consuming and computationally demanding and cause some delay. The real-time calculation is not convenient for online transmission. It would be an advantage with video on-demand, games, and virtual reality.

The presented model achieved excellent results. Each simulation was statistically verified to predict the simulated variables with a high success rate. The proposed model offering bitstream prediction by using characteristics of video sequences in the form of SI and TI information, the prediction of video quality, and the mapping of subjective quality through an objective evaluation is unique and brings added value to the field of video quality evaluation. The paper describes a full methodology leading to the fulfillment of its goals. Section 3 describes the current situation in this area and presents some of the scientific works addressing it.

A database of video sequences has been created to ensure a wide variety of possible scenarios. It contains motion pictures encoded based on different qualitative parameters with various bitstreams. These videos were subjectively evaluated by numerous observers using the ACR metric and also objectively assessed using the SSIM metric. After that, the correlation between these two evaluations was found; see in Section 7. The scene classifier is described in Section 6. The mapping function which determines the appropriate bitrate for the specified quality ratings in the form of MOS or SSIM based on the type of sequence characterized by SI and TI is described in Section 8. The next section deals with verifying and validating the proposed model, where the predicted value is compared with the reference using different statistical methods. The created model can be used in practice and implemented in real-time. This paper provides a new approach which permits adaptive reservation of network resources according to the scene.

The proposed model is able to map objective quality to subjective; and predict the subjective or objective quality of video sequences from SI, TI, and qualitative parameters. The key part is the bitrate prediction based on the requirements for the quality. All types of prediction were carried out using artificial intelligence in MATLAB. We are considering the implementation of our own solution of the neural network for improving the model and the results. Another idea is to extend our model using the next compression standards, resolutions, and bitrates. Future work will concentrate on parsing a video into small parts, and tagging them based on each scene’s type computed from SI and TI. Consequently, each segment will be assigned an appropriate bitrate. The desired quality will thus be maintained throughout the whole video sequence.

## Figures and Tables

**Figure 1 sensors-21-01949-f001:**
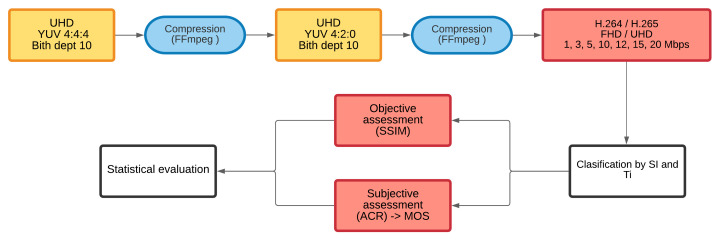
The process of coding, classification, and evaluation of quality.

**Figure 2 sensors-21-01949-f002:**
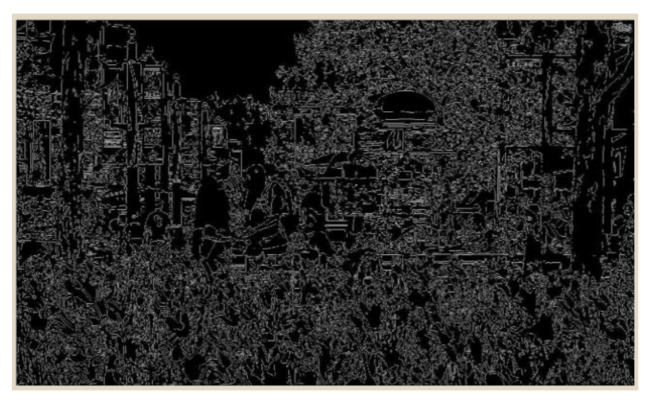
Spatial information (SI).

**Figure 3 sensors-21-01949-f003:**
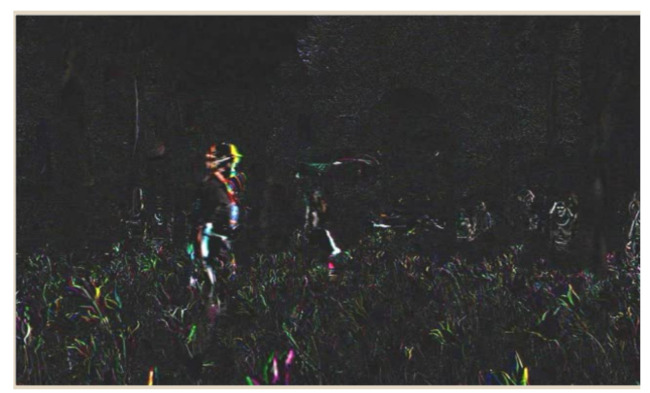
Temporal information (TI).

**Figure 4 sensors-21-01949-f004:**
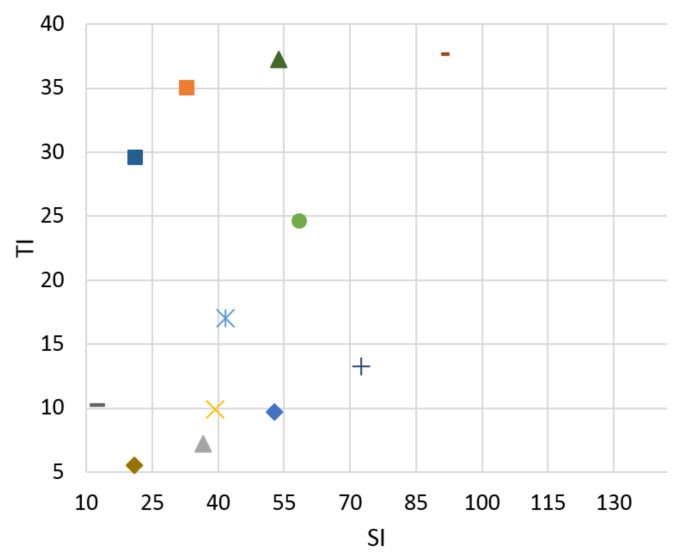
Ultra HD, H.265, bitrate 3 Mbps.

**Figure 5 sensors-21-01949-f005:**
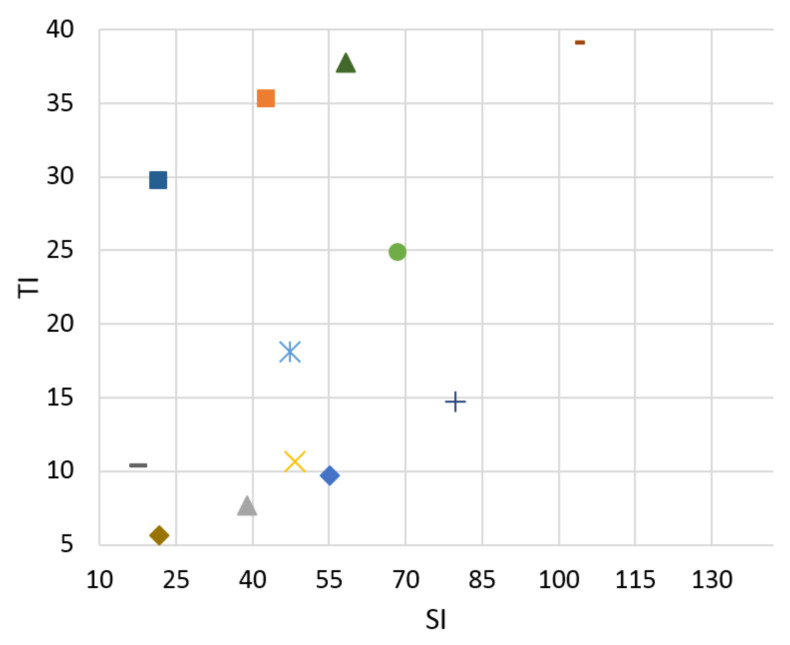
Ultra HD, H.265, bitrate 20 Mbps.

**Figure 6 sensors-21-01949-f006:**
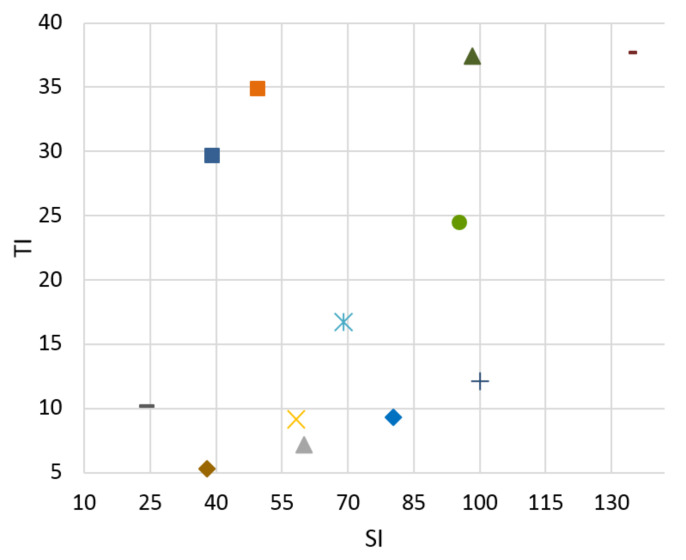
Full HD, H.265, bitrate 3 Mbps.

**Figure 7 sensors-21-01949-f007:**
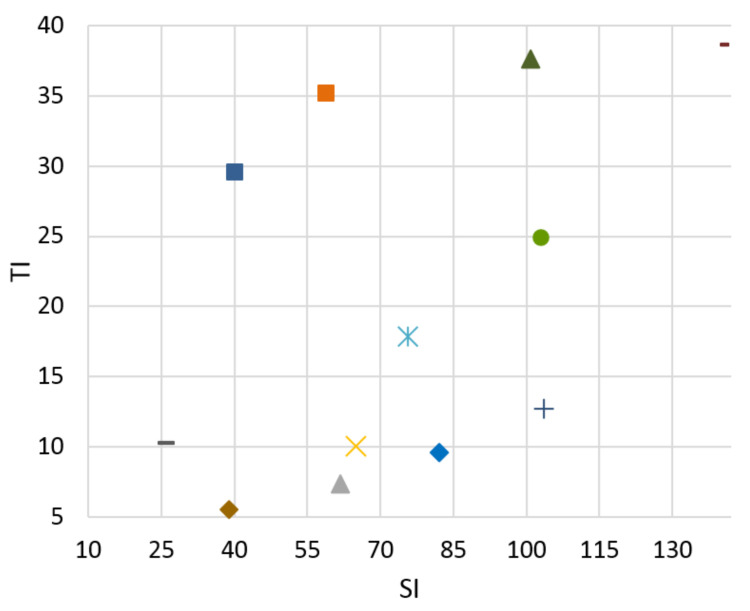
FULL HD, H.265, bitrate 20 Mbps.

**Figure 8 sensors-21-01949-f008:**
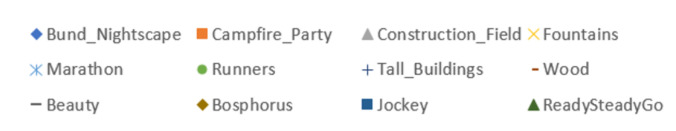
The legend of Figure 4, Figure 5, Figure 6 and Figure 7.

**Figure 9 sensors-21-01949-f009:**
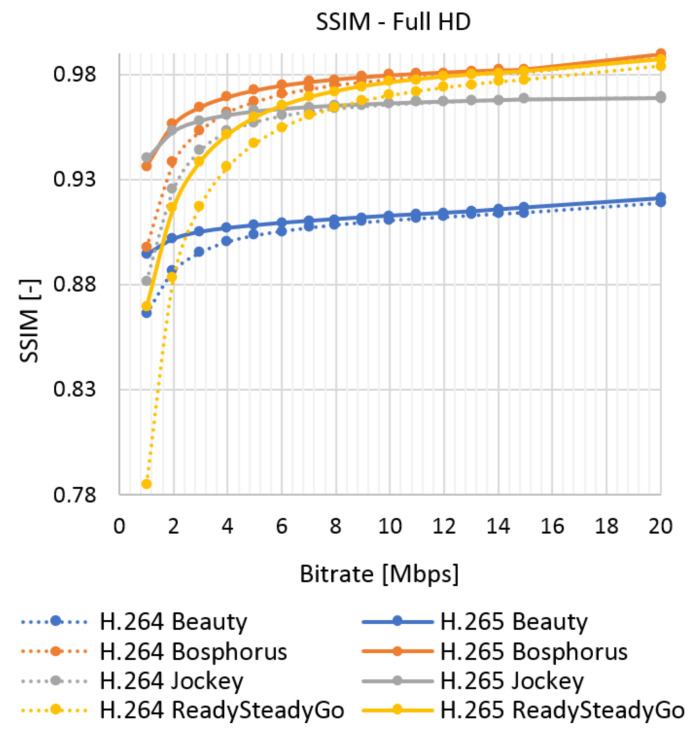
SSIM evaluation of Ultra Video Group (UVG) sequences—Full HD.

**Figure 10 sensors-21-01949-f010:**
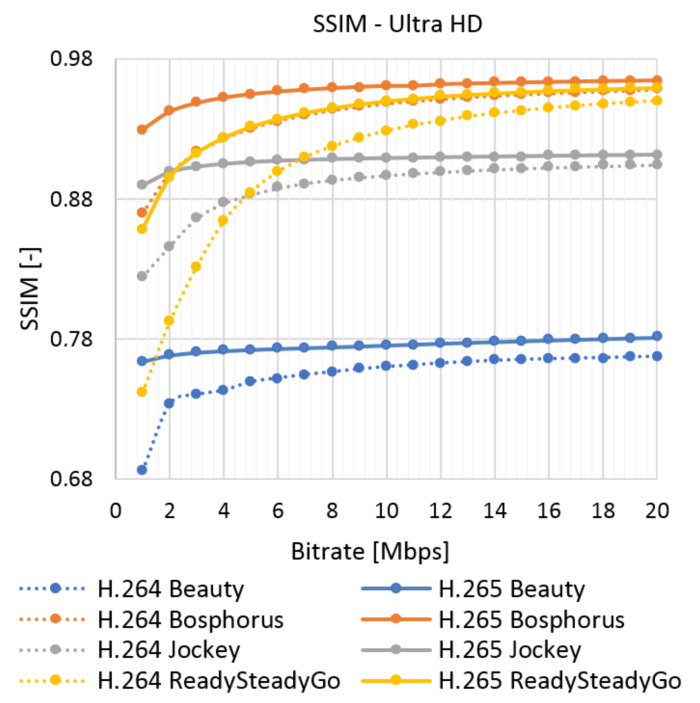
SSIM evaluation of UVG sequences—Ultra HD.

**Figure 11 sensors-21-01949-f011:**
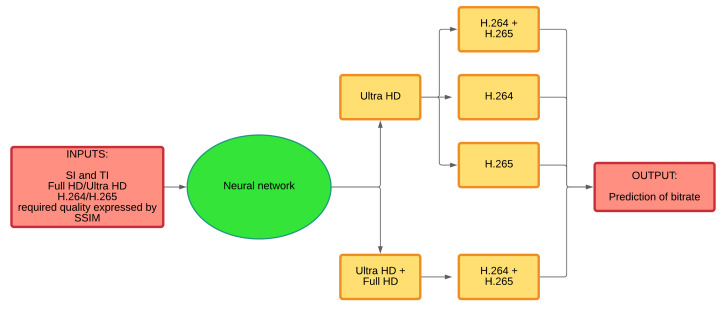
An example of the neural network.

**Figure 12 sensors-21-01949-f012:**
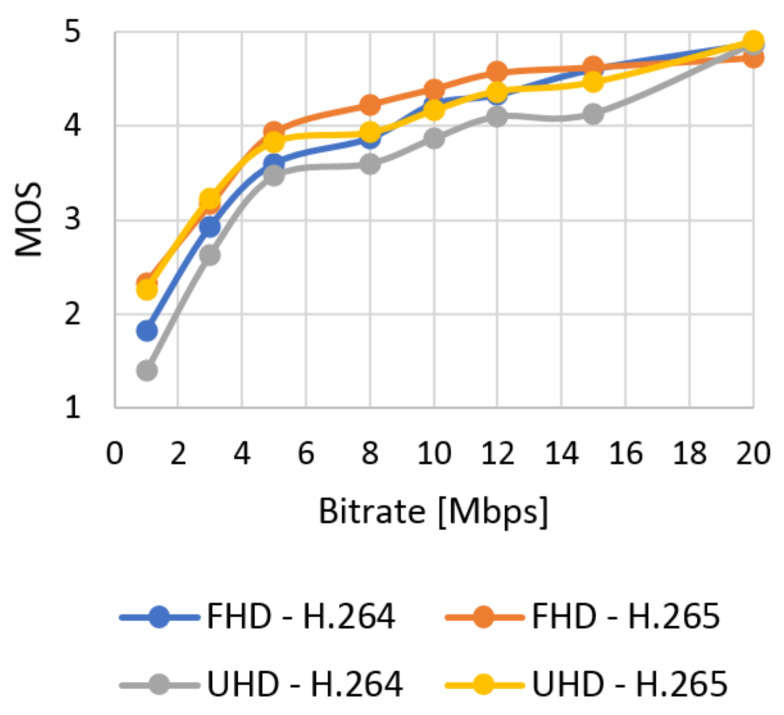
Bund Nightscape, subjective evaluation.

**Figure 13 sensors-21-01949-f013:**
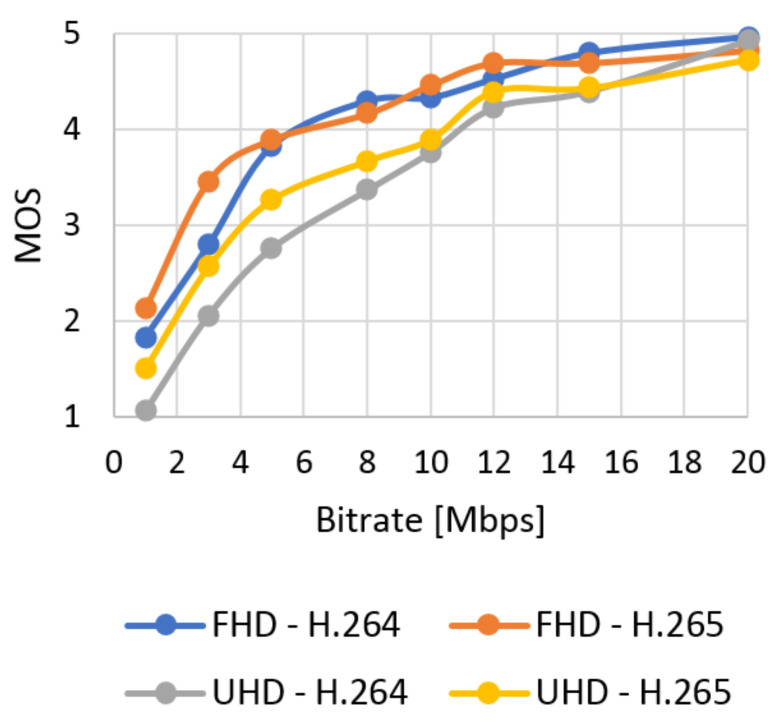
Wood, subjective evaluation.

**Figure 14 sensors-21-01949-f014:**
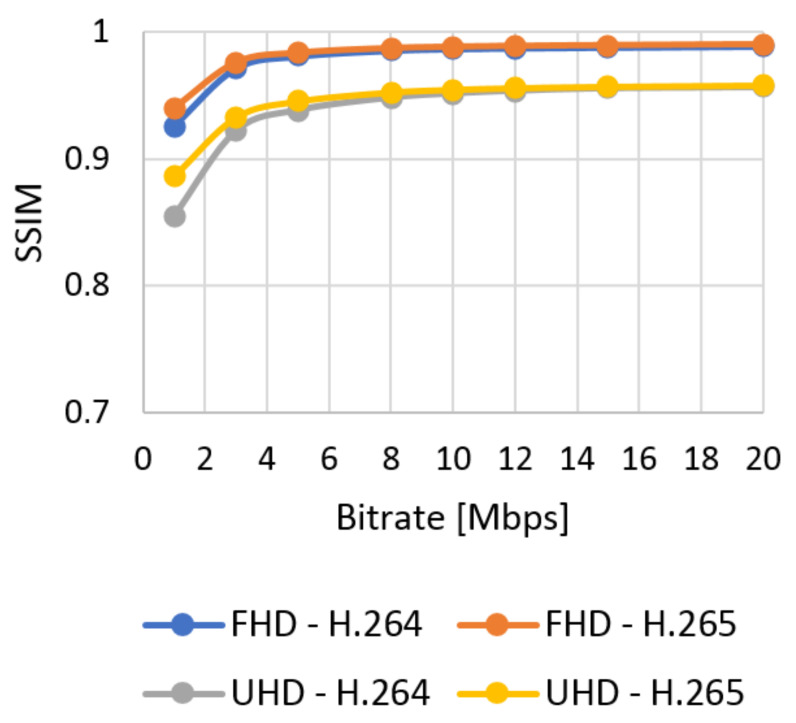
Bund Nightscape, objective evaluation.

**Figure 15 sensors-21-01949-f015:**
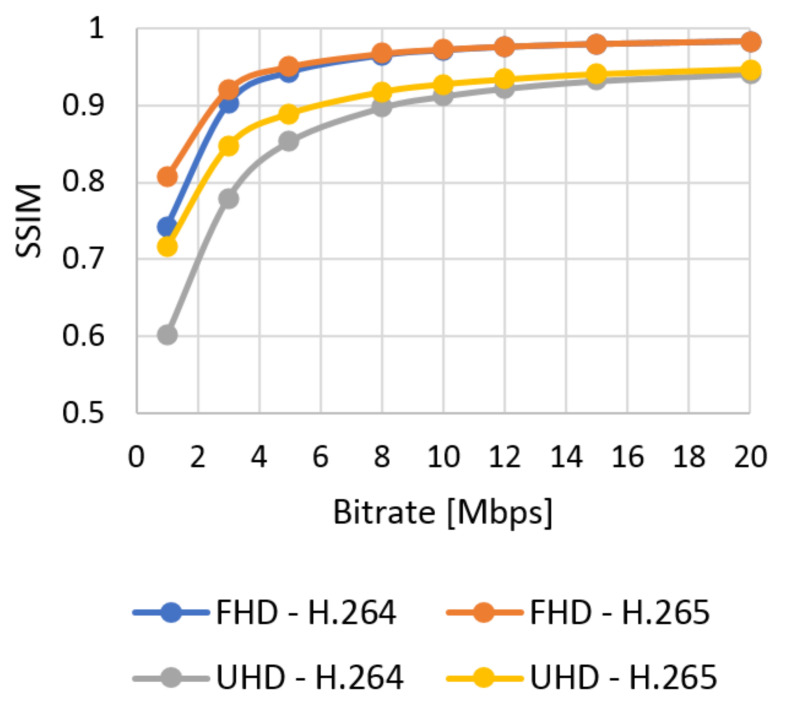
Wood, objective evaluation.

**Figure 16 sensors-21-01949-f016:**
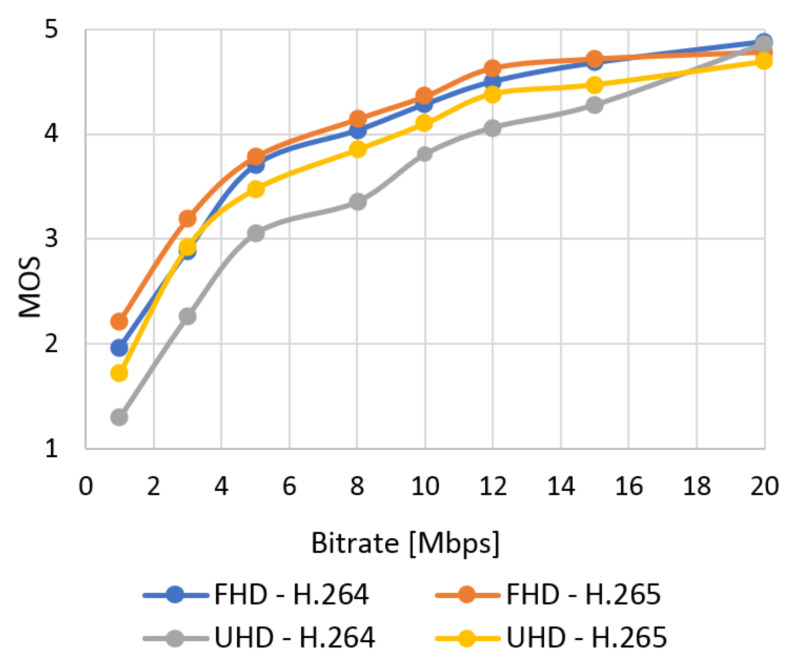
The average results of MOS.

**Figure 17 sensors-21-01949-f017:**
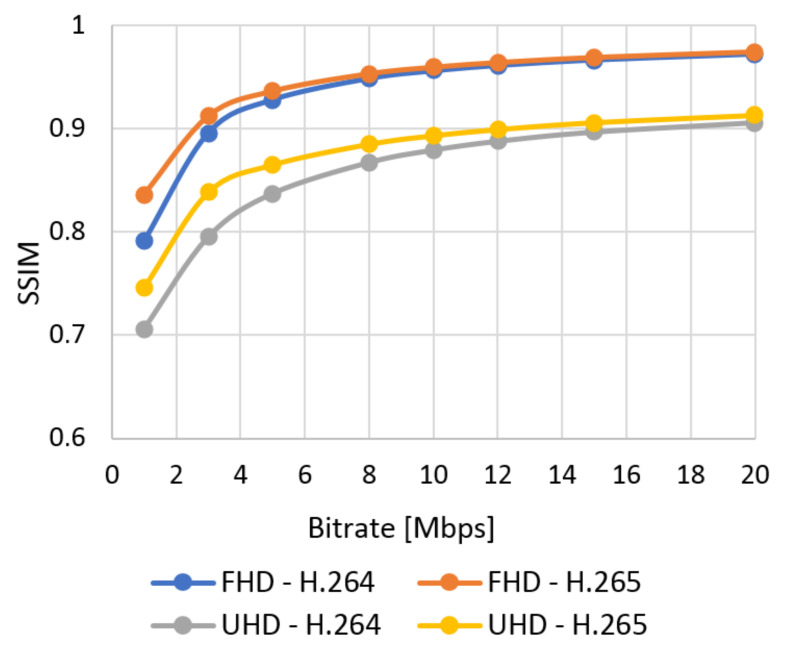
The average results of SSIM.

**Figure 18 sensors-21-01949-f018:**
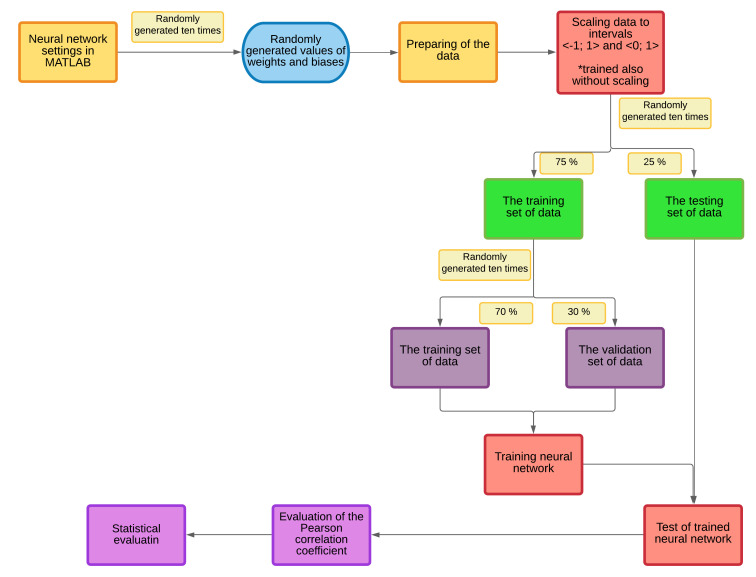
The process of creating the neural network.

**Figure 19 sensors-21-01949-f019:**
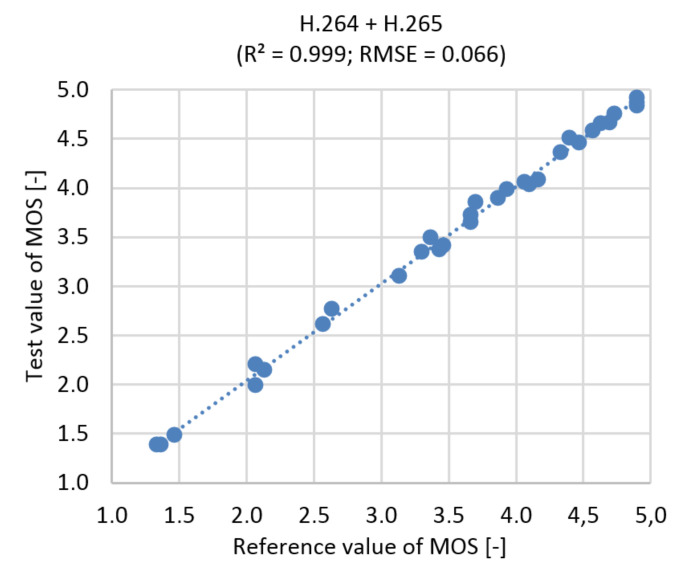
Correlation diagram for MOS prediction, Ultra HD, H.264 + H.265.

**Figure 20 sensors-21-01949-f020:**
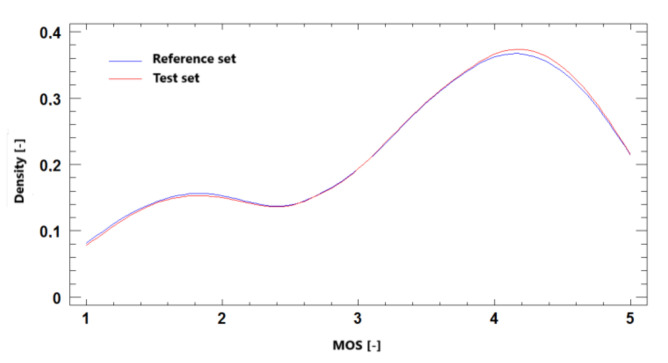
The probability density function for MOS prediction, Ultra HD, H.264 + H.265.

**Figure 21 sensors-21-01949-f021:**
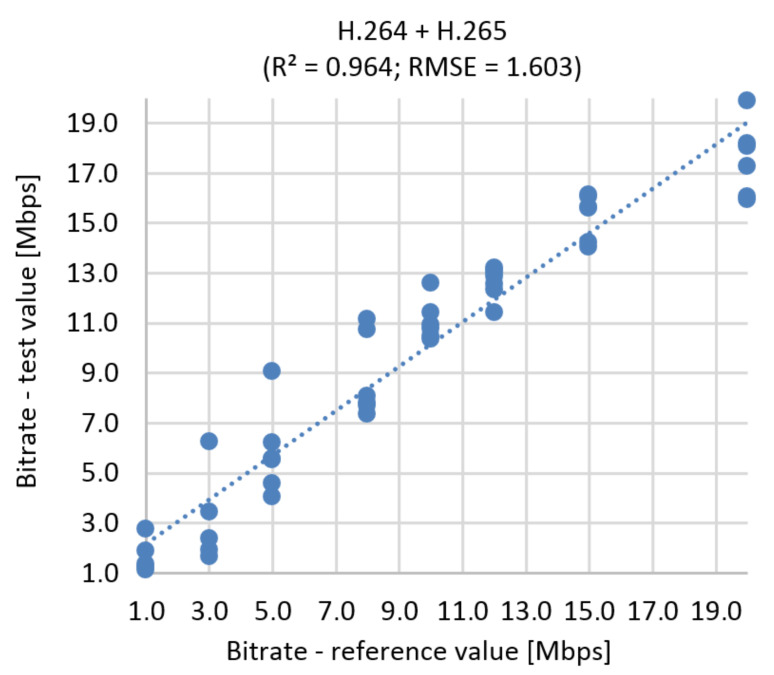
Correlation diagram for bitrate prediction based on SSIM, **Ultra HD**, H.264 + H.265.

**Figure 22 sensors-21-01949-f022:**
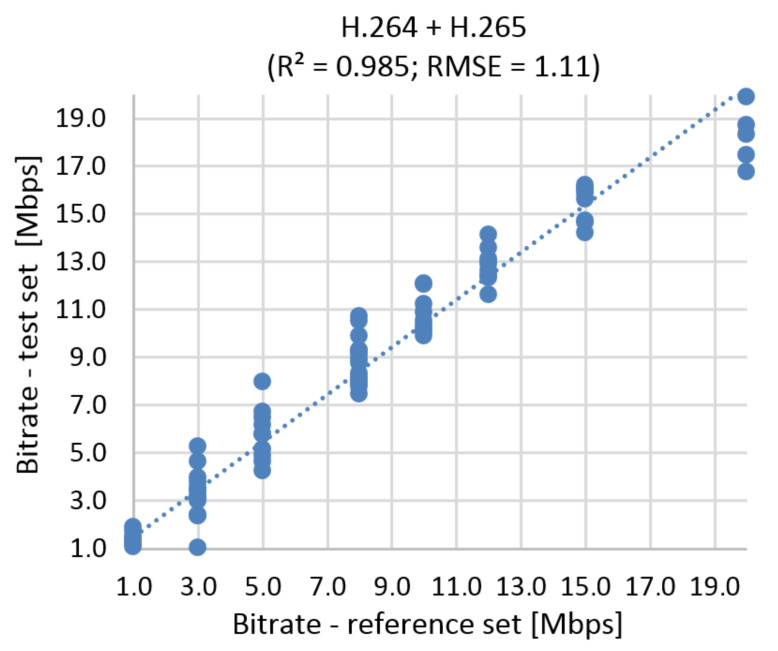
Correlation diagram for bitrate prediction based on SSIM, **Full HD + Ultra HD**, H.264 + H.265.

**Figure 23 sensors-21-01949-f023:**
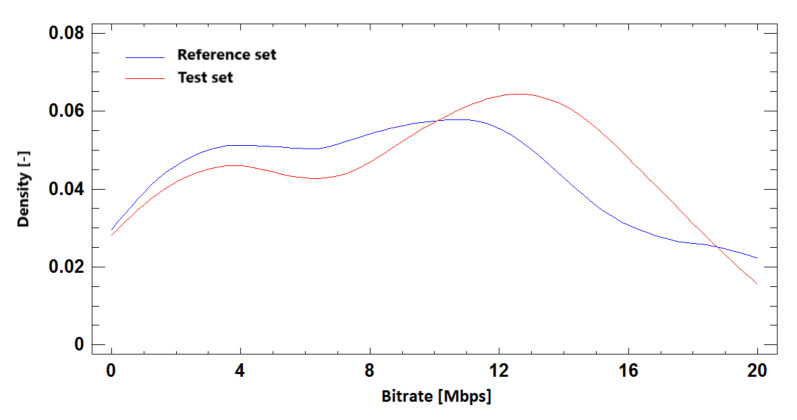
The probability density function for bitrate prediction based on SSIM, **Ultra HD**, H.264 + H.265.

**Figure 24 sensors-21-01949-f024:**
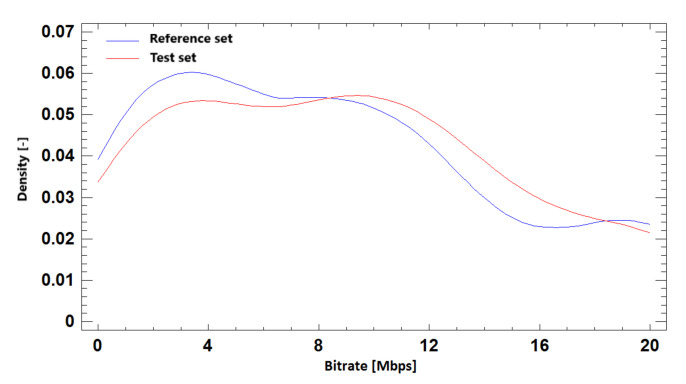
The probability density function for bitrate prediction based on SSIM, **Full HD + Ultra HD**, H.264 + H.265.

**Figure 25 sensors-21-01949-f025:**
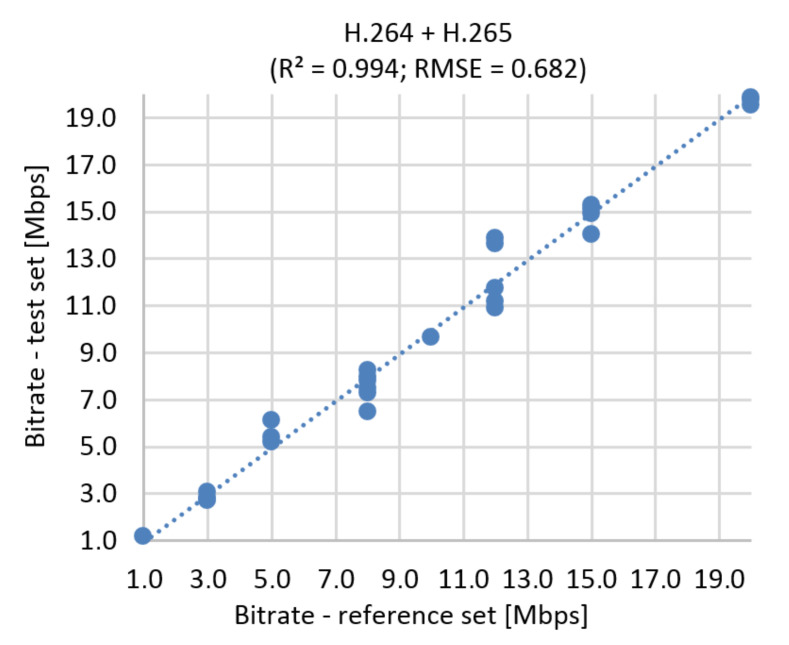
Correlation diagram for bitrate prediction based on MOS, Ultra HD, H.264 + H.265.

**Figure 26 sensors-21-01949-f026:**
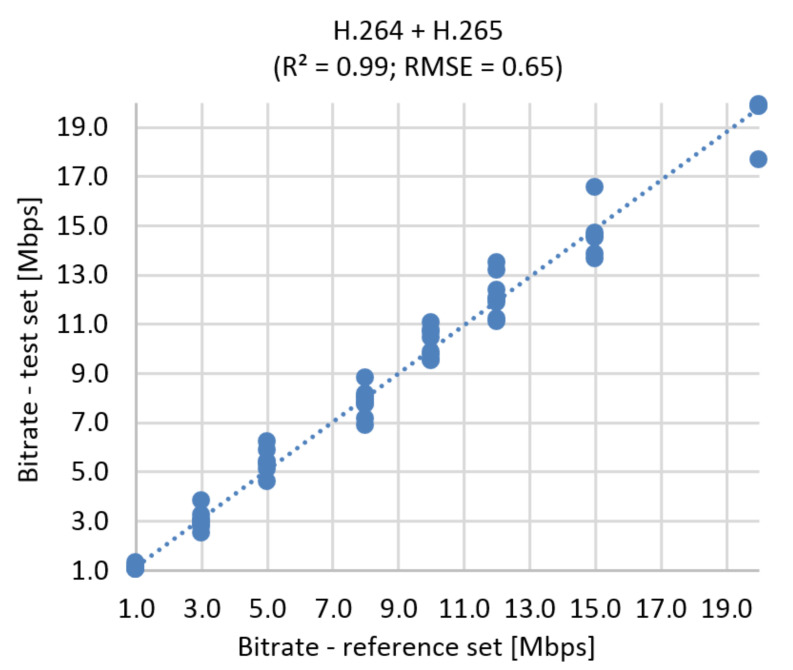
Correlation diagram for bitrate prediction based on MOS, Full HD + Ultra HD, H.264 + H.265.

**Figure 27 sensors-21-01949-f027:**
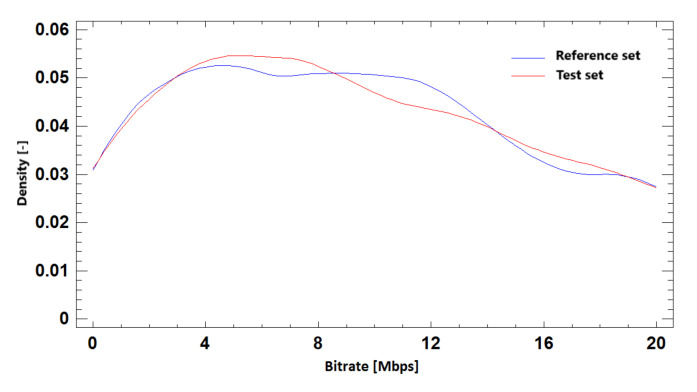
The probability density function for bitrate prediction based on MOS, Ultra HD, H.264 + H.265.

**Figure 28 sensors-21-01949-f028:**
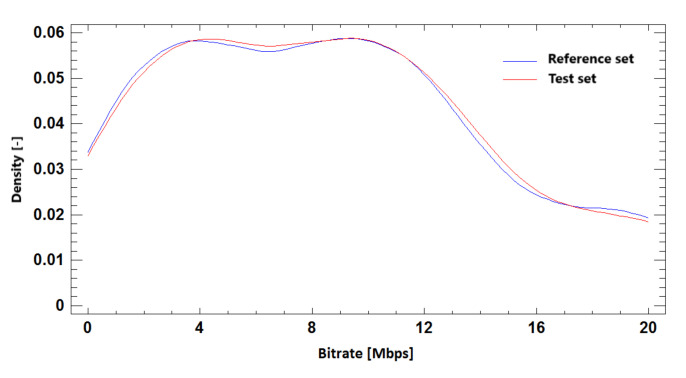
The probability density function for bitrate prediction based on MOS, Full HD + Ultra HD, H.264 + H.265.

**Table 1 sensors-21-01949-t001:** A comparison table of the selected relevant works.

	Our Paper	Paper [29]	Paper [30]
**The main idea of the paper**	Setting an appropriate bitrate based on a new classifier for predicting the boundaries SI and TI followed the quality requirements	The method of estimating the perceived quality of experience of users of UHD video flows in the emerging 5G networks is presented	The paper describes streaming of the video sequences using both the classical and the adaptive streaming approach; 3 type of Bandwidth scenarios
**Source of the video sequences**	Media Lab (8)/Ultra Video Group (4)	Ultra Video Group (4)/Mitch Martinez (5)	N/A
**Number of video sequences**	12	9	5
**Duration of the video sequences [sec.]**	10	10	60
**Definition of the individual video sequences**	SI, TI, qualitative parameters	N/A	SI, TI
**Software for the encoding**	FFmpeg	A modified version of the scalable HEVC reference software (version SHM 6.1)	N/A
**Compression standards**	H.264, H.265	H.265	N/A
**The resolution**	Full HD, Ultra HD	Full HD, Ultra HD	Quad HD
**Frame rate**	30	30/24	30/24
**Subjective evaluators**	30	64	40
**Methods of the evaluation**	Subjective (ACR), Objective (SSIM)	Subjective (ACR)	Subjective (ACR)
**Performance**	Each simulation is statistically verified with a high success rate of predicting the simulated variables	The results of subjective testing achieve an accuracy of up to 94%. MOS scores of the test subject have the maximum variance of 0.06 and 0.17.	The adaptive streaming case outperforms the standard for all the scenarios

**Table 2 sensors-21-01949-t002:** Parameters for encoding.

Parameter	Description
Resolution	Full HD, Ultra HD
Type of codec	H.264 (AVC), H.265 (HEVC)
Bitrate [Mb/s]	1, 3, 5, 8, 10, 12, 15, 20
Chroma subsampling	4:2:0
Bit depth	10
Framerate	30 fps
Duration	10 s

**Table 3 sensors-21-01949-t003:** Equation parameters.

Parameter	Description
$max_{time}$	The maximum value in the time
$std_{space}$	Standard deviation over pixels
Sobel	The Sobel filter
$f_{n}$	Number of frames in time n

**Table 4 sensors-21-01949-t004:** The basic and the best topologies of the subjective prediction; Ultra HD.

Codec	Topology	$R_{\mathrm{TRAIN}}^{2}$[-]	Time [s]	Num. of Epochs [-]
**H.264 + H.265**	5-1	0.97	64.962	681
	**51-25**	**0.996**	**12.446**	**302**
	5-1-1	0.974	99.480	762
	5-3-1	0.977	108.160	965
	5-3-2-1	0.974	91.926	820
	5-5-3-1	0.98	106.551	924
**H.264**	5-1	0.934	93.305	853
	**31-15**	**0.993**	**281.610**	**273**
	5-1-1	0.953	91.826	963
	5-3-1	0.934	135.354	1101
	5-3-2-1	0.957	152.567	1118
	5-5-3-1	0.953	94.658	888
**H.265**	5-1	0.971	96.904	808
	**39-19**	**0.992**	**30.651**	**278**
	5-1-1	0.969	91.443	836
	5-3-1	0.975	116.228	1037
	5-3-2-1	0.984	103.205	827
	5-5-3-1	0.989	118.936	918

**Table 5 sensors-21-01949-t005:** Simulation of the MOS prediction for the best topologies.

Resolution	Codec	Topology	MSE	$R_{\mathrm{TRAIN}}^{2}$[-]	$R_{\mathrm{VAL}}^{2}$ [-]	$R_{\mathrm{TEST}}^{2}$[-]
**Ultra HD**	**H.264 + H.265**	51-25	0.002	0.996	0.996	0.997
	**H.264**	31-15	0.004	0.994	0.987	0.999
	**H.265**	39-19	0.004	0.993	0.998	0.997
**Full HD+** **Ultra HD**	**H.264 + H.265**	47-23	0.006	0.986	0.991	0.993

**Table 6 sensors-21-01949-t006:** Simulation of bitrate prediction based on SSIM for the best topologies.

Resolution	Codec	Topology	MSE	$R_{\mathrm{TRAIN}}^{2}$[-]	$R_{\mathrm{VAL}}^{2}$[-]	$R_{\mathrm{TEST}}^{2}$[-]
**Ultra HD**	**H.264 + H.265**	43-21	0.041	0.949	0.944	0.963
	**H.264**	79-39	0.0001	0.999	0.999	0.999
	**H.265**	63-31	0.001	0.999	0.999	0.999
**Full HD+** **Ultra HD**	**H.264 + H.265**	47-23	0.029	0.963	0.957	0.985

**Table 7 sensors-21-01949-t007:** The best topologies of bitrate prediction based on SSIM.

Resolution	Codec	Topology	$R_{\mathrm{TEST}}^{2}$[-]	Time[s]	Num. of Epochs[-]
**Ultra HD**	**H.264 + H.265**	43-21	0.946	191.265	338
	**H.264**	79-39	0.970	363.562	266
	**H.265**	63-31	0.939	627.246	360
**Full HD + Ultra HD**	**H.264 + H.265**	47-23	0.965	217.427	294

**Table 8 sensors-21-01949-t008:** The best topologies of bitrate prediction based on MOS.

Resolution	Codec	Topology	$R_{\mathrm{TEST}}^{2}$[-]	Time[s]	Num. of Epochs [-]
**Ultra HD**	**H.264 + H.265**	47-23	0.988	12.163	270
	**H.264**	35-17	0.992	18.957	330
	**H.265**	71-35	0.990	20.299	347
**Full HD + Ultra HD**	**H.264 + H.265**	71-35	0.990	29.113	250

**Table 9 sensors-21-01949-t009:** Simulation of bitrate prediction based on MOS for the best topologies.

Resolution	Codec	Topology	MSE	$R_{\mathrm{TRAIN}}^{2}$[-]	$R_{\mathrm{VAL}}^{2}$[-]	$R_{\mathrm{TEST}}^{2}$[-]
**Ultra HD**	**H.264 + H.265**	47-23	0.013	0.982	0.991	0.992
	**H.264**	35-17	0.006	0.992	0.991	0.996
	**H.265**	71-35	0.011	0.988	0.982	0.993
**Full HD+** **Ultra HD**	**H.264 + H.265**	71-35	0.008	0.989	0.99	0.995

**Table 10 sensors-21-01949-t010:** Descriptive statistics for MOS prediction, Ultra HD, H.264 + H.265.

	Reference Set	Test Set
**Average value [-]**	3.481	3.501
**Minimum [-]**	1.333	1.382
**Maximum [-]**	4.900	4.920
**Median [-]**	3.700	3.850
**Confidence interval [-]**	<3.067; 3.896>	<3.091; 3.910>
**Pearson’s correlation coef. [-]**	0.999	
**Spearman’s correlation coef. [-]**	0.976	

**Table 11 sensors-21-01949-t011:** Descriptive statistics for bitrate prediction based on SSIM, H.264 + H.265.

	Ultra HD		Full HD + Ultra HD	
	Reference Set	Test Set	Reference Set	Test Set
**Average value [-]**	9.438	9.664	8.583	9.14
**Minimum [-]**	1.000	0.369	1	0.73
**Maximum [-]**	20.000	19.884	20.00	22.11
**Median [-]**	10.000	10.744	8.00	22.84
**Confidence interval [-]**	<7.708; 11.167>	<8.071; 11.257>	<5.43; 7.22>	<5.39; 7.18>
**Pearson’s correlation coef. [-]**	0.964		0.985	
**Spearman’s correlation coef. [-]**	0.976		0.970	

**Table 12 sensors-21-01949-t012:** Descriptive statistics for bitrate prediction based on MOS, Ultra HD, H.264 + H.265.

	Ultra HD		Full HD + Ultra HD	
	Reference Set	Test Set	Reference Set	Test Set
**Average value [-]**	9.656	9.594	8.73	8.74
**Minimum [-]**	1	0.667	1.00	0.93
**Maximum [-]**	20	20.467	20.00	19.83
**Median [-]**	8	8.106	8.00	8.05
**Confidence interval [-]**	<7.380; 11.932>	<7.307; 11.881>	<7.28; 10.19>	<7.31; 10.18>
**Pearson’s correlation coef. [-]**	0.994		0.99	
**Spearman’s correlation coef. [-]**	0.99		0.99	

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
