# Peer review of "Adaptive Reservation of Network Resources According to Video Classification Scenes"

_sensors, 2021, doi:10.3390/s21061949_

Round 1

Reviewer 1 Report

The paper presents an impressive introduction based on a great number of references, it can observe the research activities of authors of papers which show a long experience in the domain.

The methods applied are sustained by analytic demonstration with formulas. 

In conclusion, can be seeing a lot of experiments made for the implemented solution and the results present very good the analyze of the case study.

Author Response

Dear reviewer, responses to all comments are attached in separated PDF.

Reviewer 2 Report

The paper deals with the approach of mapping QoS (Quality of Service) to QoE (Quality of Experience) using QoE metrics to determine user satisfaction limits and applying QoS tools to provide the minimum QoE expected by users. 

In Abstract, research motivation should be briefly discussed. A brief conclusion needs to be provided as well.

I. Introduction should be revised. One paragraph is one section. The introduction should discuss research background, motivation, research gap, and contribution of the work. 

The literature review does not contribute much to the overall community of exploring video quality. The authors have mentioned existing works, but no new insights are provided. I request the authors to be precise and clear about how the proposed work is different from the existing relevant works. A comparison table might be needed. 

Contributions should be presented using bullet format.

Before introducing the proposed work, an overview should be provided so that the readers will have a big picture of the approach. 

Why the creation of a video database is not clear.

A background regarding neural network should be provided. 

Why the video from Shanghai Jiao Tong University and Tempa Univ are selected should be explained. 

The proposed approach should be compared with existing work to show the potential improvement. 

A flowchart might be helpful to explain each sub-step. 

At the end, a discussion section talking about design feature, potential constraints, and possible improvement should be provided. 

Author Response

(The authors gave the same response as above.)

Reviewer 3 Report

The paper is well written with a good balance between theoretical and experiments parts.

It would be best for a reader if the authors can add a block/logic diagram of the model.

The authors should state clearly what are some use-cases that this model can be used at and what are the clear benefits inside that use-cases.

Author Response

(The authors gave the same response as above.)

Round 2

Reviewer 2 Report

Can be accepted.